# Case Report of Neonatal Sotos Syndrome with a New Missense Mutation in the *NSD1* Gene and Literature Analysis in the Chinese Han Population

**DOI:** 10.3390/medicina58070968

**Published:** 2022-07-21

**Authors:** Hui-Ying Jin, Hai-Feng Li, Jia-Lu Xu, Wang Hui, Wen-Cong Ruan, Cheng-Cheng Lv, Ren-Ai Xu, Shu Qiang

**Affiliations:** Department of Rehabilitation, The Children’s Hospital, Zhejiang University School of Medicine, National Clinical Research Center for Child Health, Hangzhou 310052, China; 6507079@zju.edu.cn (H.-Y.J.); 6199005@zju.edu.cn (H.-F.L.); dew2020@zju.edu.cn (J.-L.X.); 6201038@zju.edu.cn (W.H.); 6512060@zju.edu.cn (W.-C.R.); 18767167798@163.com (C.-C.L.); 565706549@163.com (R.-A.X.)

**Keywords:** neonatal Sotos syndrome, *NSD1*, missense variation, c.5000C>A, chr5:176687023

## Abstract

Currently, no consensus exists regarding Sotos syndrome in the Chinese population. Here, we present a case of neonatal Sotos syndrome, followed by a retrospective analysis of five cases of neonatal Sotos syndrome, reported in China. The study subject was a twin premature infant, heavier than gestational age, with characteristic facial features, limb shaking, and hypertonia. Transient hypoglycemia, abnormal cranial magnetic resonance imaging, multiple nodules in polycystic kidneys and liver, abnormal hearing, patent ductus arteriosus, and an atrial septal defect were also noted. The subject showed overgrowth and developmental retardation at 3 months of age. Sequencing revealed a novel missense mutation, c.5000C>A, in the nuclear receptor binding the SET domain protein 1 gene, resulting in an alanine-to-glutamate substitution. The bioinformatics analysis suggested high pathogenicity at this site. This study provides insights into diagnosis of neonatal Sotos syndrome based on specific phenotypes. Subsequent treatment and follow-up should focus on developmental retardation, epilepsy, and scoliosis.

## 1. Introduction

Sotos syndrome is a rare overgrowth disease that shows autosomal dominant inheritance; its global incidence rate is approximately 1:14,000 [1]. Currently, limited information is available regarding neonatal Sotos syndrome in China, as only four cases have been reported [2,3,4]. Here, we present a case of neonatal Sotos syndrome diagnosed and followed up at Children’s Hospital affiliated to Zhejiang University School of Medicine (Hangzhou, China) in November 2021.

## 2. Case Presentation

### 2.1. Clinical Data

The patient was male, born on 23 November 2021, at a local hospital, and admitted to Children’s Hospital affiliated to Zhejiang University School of Medicine on 26 November 2021, because of “limb shaking for two days”. At presentation, his limbs trembled and his head swayed occasionally without any obvious stimulus. The shaking bouts lasted for approximately 10 s, followed by relief, and occurred several times a day. The patient’s blood glucose level was 1.8 mmol/L (2.6–6.5 mmol/L) but reached a normal range after the intravenous administration of glucose; however, the shaking of the limbs was not significantly relieved. From the disease onset, milk intake and urine and stool output were normal. The child was one of twin test-tube babies, born prematurely at 35 weeks and 6 days, with a birth weight of 3150 g. His Apgar score was 8 points at 1 min and 10 points at 5 min. There was no history of birth asphyxia. No significant genetic history was noted within three generations of the family. A physical examination revealed the following characteristics: head circumference 33 cm (50–90th percentile), body length 50 cm (90th percentile), weight 2770 g (50–90th percentile), irritability, frequent limb shaking, mild yellow skin, prominent forehead, mandibular tip, and increased muscle tension in the limbs. The patient’s twin sister was healthy.

### 2.2. Supplementary Examination

Upon admission, the child’s blood ammonia level was 73 µmol/L (18–72 µmol/L), and the blood, electrolyte, and cerebrospinal fluid tests and metabolic analyses revealed no obvious abnormalities. A standard electroencephalogram (EEG) showed no obvious abnormalities. The low bound voltage of the amplitude-integrated EEG of the patient was lower compared to a normal infant with the same gestational age. A discontinuous wave was seen during the quiet sleep period, and the maximum IBI was 15 µV. The echocardiography showed an atrial septal defect (3.8 mm secondary hole), patent ductus arteriosus (1.9 mm), and mild regurgitation of blood around the tricuspid valve. Magnetic resonance (MR) urography (MRU) suggested bilateral infantile polycystic kidney, bilateral testis coleoplasty, left inguinal hernia, and multiple nodules on the liver (Figure 1). Two automatic auditory brainstem-response tests showed failure at 5 and 16 days after birth. At 1 month and 17 days after birth, the auditory brainstem V-wave response threshold was 50 dbnhl in the left ear and 70 dbnhl in the right ear. At 5 days after birth, cranial magnetic resonance imaging (MRI) indicated signs of premature birth in the brain and suspected hypoplasia. It was recommended to follow up and observe the patient’s brain development (Figure 2). At 2 months and 25 days after birth, a cranial MRI showed that the extracerebral space was widened, the bilateral lateral ventricles were enlarged, the occipital cistern was large, and the myelination of brain white matter was backward (Figure 3).

After the guardians of the child provided signed informed consent, 2 mL peripheral blood from the child and his parents was sampled in ethylenediaminetetraacetic acid as an anticoagulant and genomic DNA was extracted. The whole exon region was captured, enriched using sequence capture technology (SureSelect Series Kit, Agilent, Santa Clara, CA, USA), and sequenced with a high-flux system (pe150, Illumina, San Diego, CA, USA). The sequencing results were subjected to bioinformatics and familial genetics analyses. The tests were conducted in trio mode (i.e., the DNA of the proband and his parents was sequenced in the second generation). The reported high-quality mutation sites were not verified using Sanger sequencing, whereas the reported low-quality mutation sites were verified using Sanger sequencing.

The affected child was heterozygous for a c.5000C>A (p.A1667E) variant in the *NSD1* gene located at chr5:176687023 (Table 1 and Table 2). The OMIM database showed that the *NSD1* gene is associated with Sotos syndrome type 1 (MIM#117550), which is inherited in an autosomal dominant pattern, with heterozygous mutations. The mutation was not detected in the father or mother of the affected child, which suggested that the *NSD1* gene variant in the present case was de novo.

### 2.3. Treatment Process and Follow-Up

At 27 days after birth, the child’s symptoms improved and he was discharged. At 3 months and 21 days after birth, he presented to the rehabilitation department of our hospital with the following physical characteristics: body length of 58 cm (>97th percentile), weight of 7500 g (>97th percentile), head circumference of 44 cm (>97th percentile), slow visual and auditory response (no response to stimulation), frequent spitting, inability to rise from a prone position, retracted thumb, and increased muscle tension in the limbs. He had a total score of 3 on the Alberta infant motor scale test (prone–supine–sitting–stationary, 1–2–0–0), with a percentage < 5%.

## 3. Discussion

Sotos syndrome is an autosomal dominant overgrowth disease first described by Juan Sotos in 1964. *NSD1* gene deletion or intragenic mutation is considered the main pathogenic cause of Sotos syndrome. There are significant differences in genetic variation among different populations, and the relevant clinical manifestations are also diverse [5]. Several cases of Sotos syndrome have been reported worldwide. As of 2021, 20 cases of Sotos syndrome were reported in mainland China, including five cases of neonatal Sotos syndrome. The neonatal cases are summarized in Table 3.

Tatton Brown et al. [6] reported that the typical facial features associated with Sotos syndrome are most recognizable at the age of 1–6 years, with macrocephaly and developmental retardation occurring at all ages. Characteristic facial features, overgrowth, and mental retardation are key to the diagnosis of Sotos syndrome. More than 90% of patients with Sotos syndrome have the above-mentioned symptoms. Further, patients with Sotos syndrome have various otolaryngological diseases including hearing loss, otitis media, and feeding difficulties [7]. As shown in Table 3, some cases of neonatal Sotos syndrome have the following clinical features: (1) symptoms of intrauterine or birth overgrowth; (2) hypoglycemia; (3) motor and mental retardation; (4) backward brain development on brain MRI examination, including widening of extracerebral space, enlargement of lateral ventricles, dysplasia of the corpus callosum, and delayed myelination; and (5) congenital developmental abnormalities in multiple systems, such as the auditory, cardiovascular, urinary, and digestive systems. The typical facial features of Sotos syndrome are not obvious in some neonatal cases. Clinically, these manifestations in newborns suggest a directional diagnosis of Sotos syndrome, which can be accurately diagnosed along with the establishment of the pathogenic genotype through whole-exon detection.

The genotypes and phenotypes of genetic diseases are known to have a certain correlation [8]. Sotos syndrome is associated with *NSD1* gene variation in the 5q35 chromosomal region. Abnormal *NSD1* gene expression caused by missense mutations, nonsense mutations, frameshift mutations, or microdeletions result in altered protein expression, caused by an insufficient haploid dose [9]. In European and American countries, approximately 10% of patients have microdeletions in the *NSD1* gene 5q35 [5]; the genotypes vary among populations. As of 2021, 20 cases of Sotos syndrome have been reported in China, including 11 cases of gene mutations (55%, 11/20, of which seven cases had a missense mutation, two cases had a nonsense mutation, and two cases had a frameshift mutation) and nine cases of chromosomal microdeletion (45%, 9/20); all of these were sporadic cases with no reports of familial association [10,11,12]. Compared with patients with Sotos syndrome carrying the microdeletion genotype, those carrying chromosomal mutations are more likely to show symptoms of overgrowth; however, mental retardation and symptoms related to the heart, kidney, and other organs are not obvious [13]. A missense mutation, c.5000C>A, was detected in the *NSD1* gene region of chr5:176687023 of the patient in this study, which resulted in an alanine-to-glutamate substitution at this site (p.A1667E). No heterozygous variation of *NSD1* c.5000C >A (p.A1667E) was detected in the parents of the child, suggesting that the reported *NSD1* gene variation was de novo. In the present study, the child had obvious symptoms of overgrowth, as well as significant mental and motor retardation, and abnormalities of the heart, kidney, liver, and other organs. The reported clinical manifestations of the cases in China and other countries differ greatly, possibly because of ethnic differences and mutations at this site.

Currently, a specific treatment for Sotos syndrome is lacking, and symptomatic supportive treatment remains the main approach in clinical practice [14]. For newborns with limb shaking and suspected convulsions, visually evoked EEG suggests epileptic discharge, which can be treated using antiepileptic drugs. The seizure phenotype of Sotos syndrome generally includes gaze seizures, anthermic tonic-clonic seizures, or thermal convulsions, and is usually well controlled by medication [15]. For children with mental and motor retardation, rehabilitation assessment and treatment should be carried out at the earliest opportunity. Approximately 30% of patients with Sotos syndrome may develop scoliosis [16]; a follow-up system should thus be established, and timely treatment should be given to those with the relevant symptoms. Further, growth and development indicators should be evaluated during follow-up; in particular, for patients with abnormal cranial MRI, EEG, hearing, cardiac, urinary, and digestive system examinations, the follow-up interval should be shortened as necessary. Additionally, intervention should be performed as early as possible to improve the quality of life and prognosis.

## 4. Conclusions

In summary, this study provides clues for diagnosing neonatal Sotos syndrome based on intrauterine and neonatal overgrowth, motor and mental retardation, persistent abnormalities on cranial imaging, and congenital dysplasia in multiple systems. Further, the c.5000C>A mutation detected through exon sequencing in the present study was the first report of its kind. Subsequent treatment and follow-up for neonatal Sotos syndrome focusing on developmental retardation, epilepsy, and scoliosis are recommended.

## Figures and Tables

**Figure 1 medicina-58-00968-f001:**
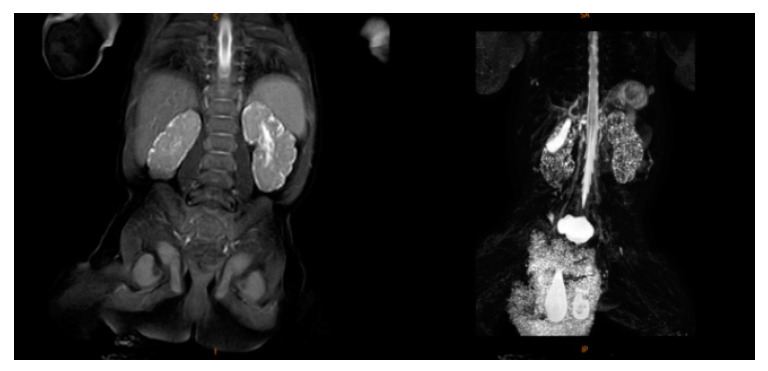
Magnetic resonance urography (MRU).

**Figure 2 medicina-58-00968-f002:**
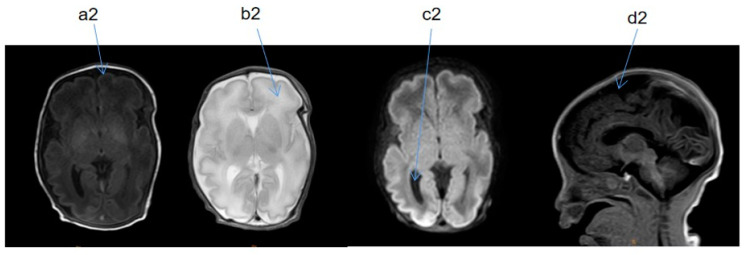
Cranial MRI taken 5 days after birth. T1 weighted imaging (TIW)—coronal section (SE), T2 weighted imaging (T2W)—SE, diffusion-weighted imaging (DWI)—SE, TIW—median sagittal section (SAG). a2: thin layer; b2: white matter has high water content. a2 and b2 indicate signs of premature birth in the brain. c2: occipital cistern was large. d2: extracranial space was widened. c2 and d2 indicate suspected brain dysplasia.

**Figure 3 medicina-58-00968-f003:**
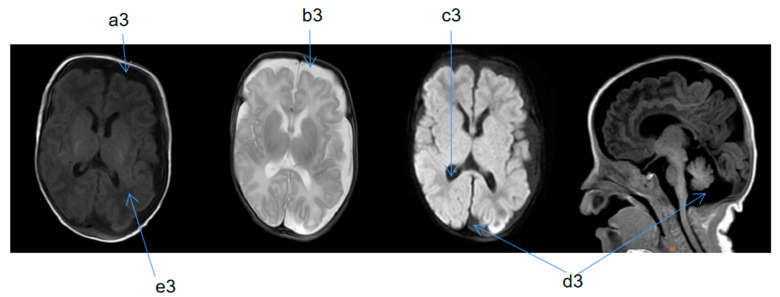
Cranial MRI taken 2 months and 25 days after birth; T1 weighted imaging (TIW)—coronal section (SE), T2 weighted imaging (T2W)—SE, diffusion-weighted imaging (DWI)—SE, TIW—median sagittal section (SAG). a3 and b3: extracerebral space was widened. c3: bilateral lateral ventricles are enlarged. d3: occipital cistern was large. e3: myelination of brain white matter was backward.

**Table 1 medicina-58-00968-t001:** Sequencing results of all exon groups (routine analysis, mainly analysis of single-nucleotide variations and indels).

Gene	Chromosome Position	Transcript Number	Exon/Intron	Nucleotide Change	Amino Acid Change	Heterozygous/ Homozygous	Related Diseases	Inheritance Mode	Variation Classification	Source of Variation
*NSD1*	chr5:176687023	NM_02245 5.4	Exon14	c.5000C>A	p.A1667E	Heterozygous	Sotos syndrome type 1	AD	Likely pathogenic	De novo mutation

Note: The reference genome version was GRCh37/hg19. AD, autosomal dominant inheritance.

**Table 2 medicina-58-00968-t002:** Statistical data of sequencing quality in the target region.

Sample Name	Sample Number	Amount of Detection Data (bp)	Average Sequencing Depth	Target Area Coverage	10× above Coverage Interval Proportion	20× above Coverage Interval Proportion
Child	WES21120171	11703902400	152.34	99.92%	99.78%	99.47%
Father	WES21120172	12428848500	162.82	99.93%	99.80%	99.50%
Mother	WES21120173	12989484300	170.16	99.80%	99.68%	99.41%

**Table 3 medicina-58-00968-t003:** Summary of five previously reported cases of neonatal Sotos syndrome in China.

	This Study	Sun, B.J. [2]		Chen, L.L. [3]	Xian, W. [4]
Case	1	2	3	4	5
Gender	Male	Male	Male	Male	Male
Age	2 days	6 h	3 days	9 days	9 days
Premature delivery	+	-	-	+	-
Time of overgrowth appearance	Intrauterine	At birth	3 months	At birth	At birth
Special facial features	+	+	−	−	+
Hypoglycemia	+	+	-	-	+
Dystonia	+	−	−	−	+
Cranial MRI abnormalities	+	+	+	+	+
Abnormal hearing	+	−	−	+	−
Abnormal EEG	+	−	−	+	−
Congenital heart defect	+	−	−	+	−
Urinary system problems	+	+	+	−	−
Developmental retardation	+	+	+	+	+
*NSD1* mutation	chr5 176687023	5q35.2–5q35.1.97 MB missing in zone 3	The long arm of chromosome 5	Yes, loci ambiguous	5q35,2q35.3 region missing

Abbreviations: +, positive: −, negative.

## Data Availability

Data are available on request from the authors.

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
