# Peer review of "Case Report of Neonatal Sotos Syndrome with a New Missense Mutation in the NSD1 Gene and Literature Analysis in the Chinese Han Population"

_medicina, 2022, doi:10.3390/medicina58070968_

Round 1
Reviewer 1 Report
Dear Author,
This paper is well written. However, it needs some corrections and additions.
Abstract
1. Since this article is a case report, the abstract part should be corrected and rewrighted according to the principles of case report abstract format
Case presentation
1. ‘‘A history of asphyxia rescue and the genetic history of his 47 family were not available. ‘’
Can you explain this sentence? Did you mean ‘uneventful’?
Supplementary examination
1. ‘‘Upon admission, the child’s blood ammonia level was 73 µmol/L (↑), and blood, elec- 54 trolyte, and cerebrospinal fluid tests and genetic and metabolic analyses revealed no ob- 55 vious abnormalities. ‘’
Could you write down the normal values of blood ammonia level for this age. Could you remove the arrow sign, (for higher values) taht is may be considered normal in newborns, in this age.
Figures
1. Could you specify the pathologies seen in MR in figures 2 and 3 in the figure legend.
Author Response
We really appreciate you for your carefulness and conscientiousness. Your suggestions are really valuable and helpful for revising and improving our paper. According to your suggestions, we have made the following revisions on this manuscript:
Abstract
- Since this article is a case report, the abstract part should be corrected and rewrighted according to the principles of case report abstract format
Response:Thank you very much for your advice,we revised it according to the format principle of case report(page 1, lines 10-22).
Case presentation
- ‘‘A history of asphyxia rescue and the genetic history of his 47 family were not available. ‘’
Can you explain this sentence? Did you mean ‘uneventful’?
Response:Thank you very much for your advice。We mean”No history of asphyxiation at birth. There was no significant genetic history within three generations of the family”(page 2, lines 44-45).
Supplementary examination
- ‘‘Upon admission, the child’s blood ammonia level was 73 µmol/L (↑), and blood, elec- 54 trolyte, and cerebrospinal fluid tests and genetic and metabolic analyses revealed no ob- 55 vious abnormalities. ‘’
Could you write down the normal values of blood ammonia level for this age. Could you remove the arrow sign, (for higher values) taht is may be considered normal in newborns, in this age.
Response:Thank you very much for your advice,The normal range of newborn blood oxygen is 18umol/L-72umol/L,We have modified it.(page 2, lines 51-52).
Figures
- Could you specify the pathologies seen in MR in figures 2 and 3 in the figure legend.
Response:Thank you very much for your advice, We have specify the pathologies seen in MR in figures 2 and 3 in the figure legend.(page 3-4, lines 74-86).
Yours sincerely,
Jin Hui Ying
Department of Rehabilitation, the Children’s Hospital, Zhejiang University School of Medicine, National Clinical Research Center for Child Health, Hangzhou, China
E-Mail:mejhy2013@163.com
Reviewer 2 Report
I thank the editors and authors for the opportunity to review this work.
The authors detail a case report of neonatal diagnosis of Sotos syndrome, and provide a review of similar published cases in China.
If Sotos syndrome is well described, it may be interesting to report possible particularities in the Chinese population. Moreover, the mutation reported here seems to be a unknown one.
I will detail the remarks following the thread of the article.
I have no remark about the abstract.
About the introduction, it could be interesting to explain "our hospital": which one, the type of center (tertiary, university?).
Li 47; about the sentence: "a history of asphyxia rescue and the genetic history of his family were not available": are the authors suggesting that there was no argument for a birth asphyxia, and that there was history of genetic disorder among the relatives?
Li 50, about abnormal "limb shaking". Can the authors clarify if these were seizures, or how these movements were different from seizures? Seizures have been reported in Sotos syndrome.
Li 55; regarding genetic and metabolic analysis: could you specify which ones?
Li 57-58, was a standard EEG performed? What do you consider as reference range about the amplitude-integrated EEG at this age?
Li 60: "tricuspid valve was slightly regurgitated": please rephrase.
Li 66: MR head plain scan: do you mean "brain MRI" ? Or "cerebral MRI" ?
Li 67: "brain hypoplasia": do you mean brain atrophy ? This requires detailing the radiological findings. Focal atrophy ? Diffuse ? White or grey substance? Please give details.
Li 69: "Morphology of bilateral ventricles was full": please rephrase.
Li 83 "After the guardians..." do you mean the parents?
Table 3:
" Age " in line 4: do you mean age at diagnosis ?
Head MRI line 9: do you mean Brain MRI findings ?
Abnormal cardiac color doppler ultrasound li 12: you could evoke congenital heart defect ?
Discussion
Li 135-138
A classic mode of revelation of sotos syndrome in the neonatal period is the association of hypoglycemia with advanced biometrics, which your case presented. It might be necessary to emphasize on this rather typical picture.
Li 146 "point mutation": please rephrase
Li 166-167: The treatments are above all symptomatic, it is perhaps not necessary to detail them?
Concerning the bibliography: it could be interesting to quote the very complete OMIM page on the subject (https://omim.org/entry/117550), as well as the publication of Tatton-Brown et al. in Gene reviews (atton-Brown K, Cole TRP, Rahman N. Sotos Syndrome. 2004 Dec 17 [Updated 2019 Aug 1]. In: Adam MP, Mirzaa GM, Pagon RA, et al. editors. GeneReviews® [Internet]. Seattle (WA): University of Washington, Seattle; 1993-2022. Available from: https://www.ncbi.nlm.nih.gov/books/NBK1479/)
Overall: what about the parents agreement to publish this case report? This is not stated in the text.
Finally, the article could benefit from correction and editing of the English.
Author Response
We really appreciate you for your carefulness and conscientiousness. Your suggestions are really valuable and helpful for revising and improving our paper. According to your suggestions, we have made the following revisions on this manuscript:
About the introduction, it could be interesting to explain "our hospital": which one, the type of center (tertiary, university?).
Response:Thank you very much for your advice,our hospital is Children's Hospital affiliated to Zhejiang University School of Medicine.We have modified it.(page 2, lines 35).
Li 47; about the sentence: "a history of asphyxia rescue and the genetic history of his family were not available": are the authors suggesting that there was no argument for a birth asphyxia, and that there was history of genetic disorder among the relatives?
Response:Thank you very much for your advice,the sentence means”No history of asphyxiation at birth. There was no significant genetic history within three generations of the family”.We have modified it.(page 2, lines 44-45).
Li 50, about abnormal "limb shaking". Can the authors clarify if these were seizures, or how these movements were different from seizures? Seizures have been reported in Sotos syndrome.
Response:Thank you very much for your advice,the abnormal "limb shaking “proved not to be seizures. Because the shaking was accompanied by no abnormal consciousness, Standard electroencephalogram(EEG) showed no obvious abnormalities(page 2, lines 53-54), and there was no recurrence after hospitalization and discharge.Seizures have been reported in Sotos syndrome. Sotos syndrome is associated with epilepsy and should be closely watched during follow-up.
Li 55; regarding genetic and metabolic analysis: could you specify which ones?
Response:Thank you very much for your advice, We make a mistake. What's right is“metabolic analyses”.We have modified it(page 2, lines 52-53).Metabolic analyses Including 47 amino acids, organic acids, fatty acids and other congenital metabolic defects, respectively:
Metabolic disease of 18 amino acids
1 carbamyl phosphate synthase deficiency
2 Ornithine carbamoyltransferase deficiency
3 Tyrosinemia Type I (Fumaryl acetyl acetyl hydrolase)
4 Tyrosinemia Type N (tyrosine aminotransferase)
5 Tyrosinemia stream type (4-hydroxyphenylpyruvate oxidase)
6 Maple diabetes mellitus (Branched A-ketoate dehydrogenase)
7 Citrullinemia Type I (arginine succinate synthase)
8 Citrate N (aspartate glutamate carrier [citric acid]) 9 Arginine succinase deficiency (arginine succinic acid lyase deficiency)
10 Hyperarginemia (arginase)
11 Homocysteine uria (Cysteine B synthase [CBS, also known as homocysteine type I)
12 Hypermethionemia (methionine adenosine cobalamin triphosphate adenosine transferase)
Histidinemia
14 ornithine 5- aminotransferase deficiency
15Nonketo hyperglycinemia
16 hyperprolinemia
17 5-hydroxyprolinemia
18Hyperornithine - hyperammonia-hyperhomoluria syndrome
Metabolic diseases of 14 organic acids
1 methylmalonic acidemia
2 propionic acidemia (propionyl-coA carboxylase)
3 isopentacidemia (isopentyl-coA dehydrogenase)
4 Glutaracidemia type I (glutaryl-coA dehydrogenase)
5 Biotinase deficiency
6 Full carboxylase synthetase deficiency
7 3-methyl-Budoloyl-coA carboxylase deficiency (3-methyl-Budoloyl-CoA carboxylase, A,B)
8 3-methylpenediacidemia (3-methylpenediacyl-coA hydrolase)
9 3-hydroxy-3-methylglutaracidemia (3-hydroxy-3-methylglutaracyl-coA lyase)
10 B ketol thiolase deficiency (B ketol thiolase)
11 2-methyl-3-hydroxybutaneemia (2-methyl-3-hydroxybutanoyl-coA dehydrogenase)
12 Malonic acidemia (Malonyl-coA decarboxylase)
13 2-methyl-butyryl-glytinuria (2-methyl-butyryl-coA dehydrogenase)
14 Isobutyl-coA dehydrogenase (ISObutyl-coA dehydrogenase)
Fatty acid oxidation defect diseases
1 Primary carnitine deficiency
2 Carnitine palmityl transferase deficiency type I (Carnitine palmityl LA)
3 Carnitine palmityl transferase deficiency n (Carnitine palmityl II)
4 Carnitine-acylcarnitine transferase deficiency (Carnitine-acylcarnitine transferase)
5 Short acyl-coA dehydrogenase deficiency (short acyl-coA dehydrogenase)
6 Medium chain acyl-coA dehydrogenase deficiency (MEDIUM chain acyl-coA dehydrogenase)
7 Extremely long chain acyl-coA dehydrogenase deficiency (extremely long chain acyl-coA dehydrogenase)
8 Short chain L-3-hydroxyl-coA dehydrogenase deficiency (SHORT chain L-3-hydroxyl-coA dehydrogenase)
9 Long chain hydroxyl-coA dehydrogenase deficiency (Long chain 3-hydroxyl-dehydrogenase)
10 Glutaracidemia Type N (electron transfer flavin protein [ETF; A,B subunit, ETFDH)
11 Trifunctional protein deficiency (trifunctional protein [A,? Subunit])
12 ethyl malonic acidemia
13 Medium chain acyl-coA thiolase deficiency (MEDIUM chain acyl-coA thiolase)
14 2, 4-dienol-coA reductase deficiency (2, 4-dienol-coA reductase)
15 Long chain acyl-coA dehydrogenase deficiency
Li 57-58, was a standard EEG performed? What do you consider as reference range about the amplitude-integrated EEG at this age?
Response:Thank you very much for your advice, We're not making the sentence clear here.We have modified it(page 2, lines 53-75).Standard EEG showed no obvious.Amplitude‑integrated electroencephalography(aEEG): the lower boundary of the aEEG of the patient was lower than that of an infant born after the same length of pregnancy. A discontinuous wave was seen during the quiet sleep period, and the maximum IBI was 15. Evaluation of brain function according to eeg background activity: namely, neonatal EEG activity is classified according to amplitude and continuity of upper and lower boundaries of aEEG. At present, the five-category method proposed by HellstromWestas et al is generally adopted.This case is mildly abnormal.
Li 60: "tricuspid valve was slightly regurgitated": please rephrase.
Response:Thank you very much for your advice,We have modified it(page 2, lines 59).
Li 66: MR head plain scan: do you mean "brain MRI" ? Or "cerebral MRI" ?
Response:Thank you very much for your advice, MR head plain scan mean “Cranial magnetic resonance imaging(MRI) ”,We have made the modification(page 2, lines 65、68).
Li 67: "brain hypoplasia": do you mean brain atrophy ? This requires detailing the radiological findings. Focal atrophy ? Diffuse ? White or grey substance? Please give details.
Response:Thank you very much for your advice, "suspected hypoplasia"(page 2, lines 66-67)means occipital cistern was large and extracranial space widened, It is marked in Figure 2(page3, lines 74-79).
Li 69: "Morphology of bilateral ventricles was full": please rephrase.
Response:Thank you very much for your advice, We have made the modification.“Bilateral lateral ventricles are enlarged”(page2, lines69).
.
Li 83 "After the guardians..." do you mean the parents?
Response:Yes,mean the parents.
Table 3:
" Age " in line 4: do you mean age at diagnosis ?
Response:" Age " in line 4: mean age of initial clinical diagnosis, not final genetic diagnosis. Because genetic testing can take some time.
Head MRI line 9: do you mean Brain MRI findings ?
Response:Yes,head MRI line 9 mean There are Cranial MRI abnormalities, We have modified it.
Abnormal cardiac color doppler ultrasound li 12: you could evoke congenital heart defect ?
Response:Thank you very much for your advice, We have modified it(Table 3。 line 13)
Discussion
Li 135-138
A classic mode of revelation of sotos syndrome in the neonatal period is the association of hypoglycemia with advanced biometrics, which your case presented. It might be necessary to emphasize on this rather typical picture.
Response:Thank you very much for your advice.We have modified it(Table 3。 line 8,page 6, line 138)
Li 146 "point mutation": please rephrase
Response:Thank you very much for your advice.Point mutations are structural changes within a gene that can be inherited.We have modified itpage7, line 152)
Li 166-167: The treatments are above all symptomatic, it is perhaps not necessary to detail them?
Response:Thank you very much for your advice.Point mutations are structural changes within a gene that can be inherited.We have modified itpage7, line 170-171)
Concerning the bibliography: it could be interesting to quote the very complete OMIM page on the subject (https://omim.org/entry/117550), as well as the publication of Tatton-Brown et al. in Gene reviews (atton-Brown K, Cole TRP, Rahman N. Sotos Syndrome. 2004 Dec 17 [Updated 2019 Aug 1]. In: Adam MP, Mirzaa GM, Pagon RA, et al. editors. GeneReviews® [Internet]. Seattle (WA): University of Washington, Seattle; 1993-2022. Available from: https://www.ncbi.nlm.nih.gov/books/NBK1479/)
Response:Thank you very much for your advice. We have modified it.
Overall: what about the parents agreement to publish this case report? This is not stated in the text.
Response:The parents have agreed to publish the case report, signed the informed consent form and uploaded it to the website as an attachment
Finally, the article could benefit from correction and editing of the English.
Response:Thank you very much for your advice. We have modified it.
Thank you again for your valuable comments and suggestions. I look forward to hearing from you soon in due course.
Yours sincerely,
Jin Hui Ying
Department of Rehabilitation, the Children’s Hospital, Zhejiang University School of Medicine, National Clinical Research Center for Child Health, Hangzhou, China
E-Mail:mejhy2013@163.com
Reviewer 3 Report
The authors report a case of a neonate with Sotos syndrome. They present history, clinical findings, results of imaging studies and genetic sequencing. Moreover, they summarize the available literature on Sotos Syndrome in the population of interest. Although the topic is of interest for the general pediatrician, an extensive editing of English language and style are required.
The following list is not exhaustive:
- Line 16. " high muscle tone" should be changed in hypertonia
- Line 43 " liquid glucose" should be changed in intravenous glucose
- Line 46 Apgar score at and not "in 1" minute, etc
- Line 47 " asphyxia rescue" should be changed in history of birth asphyxia or the baby did not require resuscitation at birth
- Line 60 " artery catheter"? did you mean ductus arteriosus or PDA
Author Response
We really appreciate you for your carefulness and conscientiousness. Your suggestions are really valuable and helpful for revising and improving our paper. According to your suggestions, we have made the following revisions on this manuscript:
- Line 16. " high muscle tone" should be changed in hypertonia
Response:Thank you very much for your advice. We have modified it(page1, line 14).
- Line 43 " liquid glucose" should be changed in intravenous glucose
Response:Thank you very much for your advice. We have modified it(page2, line 40).
- Line 46 Apgar score at and not "in 1" minute, etc
Response:Thank you very much for your advice. We have modified it(page2, line 43-44).
- Line 47 " asphyxia rescue" should be changed in history of birth asphyxia or the baby did not require resuscitation at birth
Response:Thank you very much for your advice. We have modified it(page2, line 44).
- Line 60 " artery catheter"? did you mean ductus arteriosus or PDA
Response:mean pulmonary disease anemia(PDA).We have modified it(page2, line58).
Extensive editing of English language and style required
Response:Thank you very much for your advice. We have modified it.
Thank you again for your valuable comments and suggestions. I look forward to hearing from you soon in due course.
Yours sincerely,
Jin Hui Ying
Department of Rehabilitation, the Children’s Hospital, Zhejiang University School of Medicine, National Clinical Research Center for Child Health, Hangzhou, China
E-Mail:mejhy2013@163.com
Round 2
Reviewer 3 Report
The manuscript has been improved.
However, English language and style still need to be improved. The following points are not exaustive.
Line 55-56 Change as follows: The low bound voltage of the amplitude-integrated EEG of the patient was lower compared to a normal infant with the same gestational age.
Line 59-60: I have never heard of pulmonary disease anemia, I still think that authors refer to a patent ductus arteriosus (PDA)
Author Response
We really appreciate you for your carefulness and conscientiousness. Your suggestions are really valuable and helpful for revising and improving our paper. According to your suggestions, we have made the following revisions on this manuscript:
Line 55-56 Change as follows: The low bound voltage of the amplitude-integrated EEG of the patient was lower compared to a normal infant with the same gestational age.
Response:Thank you very much for your advice. We have modified it(page2, line54-56).
Line 59-60: I have never heard of pulmonary disease anemia, I still think that authors refer to a patent ductus arteriosus (PDA)
Response:Thank you very much for your advice. We mean patent ductus arteriosus(PDA),we have modified it(page2, line58).
Thank you again for your valuable comments and suggestions. I look forward to hearing from you soon in due course.
Yours sincerely,
Jin Hui Ying
Department of Rehabilitation, the Children’s Hospital, Zhejiang University School of Medicine, National Clinical Research Center for Child Health, Hangzhou, China
E-Mail:mejhy2013@163.com